# Distinct winter North Atlantic climate responses to tropical and extratropical eruptions over the last millennium in PMIP simulations and reconstructions

Qin Tao<sup>1</sup>, Cheng Shen<sup>2</sup>, Raimund Muscheler<sup>1</sup>, Jesper Sjolte<sup>1</sup>

15

Department of Geology, Lund University, Lund, Sweden <sup>2</sup>Regional Climate Group, Department of Earth Sciences, University of Gothenburg, Gothenburg, Sweden \*Correspondence to: Jesper Sjolte (jesper.sjolte@geol.lu.se)

Abstract. Large tropical (TROP) volcanic cruptions can influence North Atlantic climate by inducing a positive shift of the North Atlantic Oscillation (NAO), typically resulting in winter warming across northern Eurasia. In contrast, Northern Hemisphere extratropical (NHET) cruptions are proposed to have opposite impacts, though uncertainties exist regarding the performance of climate models in capturing these differences. This study examines winter North Atlantic climate responses to TROP and NHET cruptions using last millennium simulations and paleoclimate reconstructions. We find distinct differences in NAO related climate responses to TROP and NHET cruptions in both simulations and reconstructions, depending on the selection for cruption events. Notably, models employing the latest volcanic forcing dataset exhibit improved agreement with paleoclimate reconstructions. These findings highlight the critical need for improved volcanic forcing datasets, refined paleoclimate reconstructions, and robust statistical approaches to better constrain uncertainties in assessing the simulated volcanic impacts on North Atlantic climate.

Abstract. Large tropical (TROP) volcanic eruptions can influence North Atlantic climate by inducing a positive shift of the 20 North Atlantic Oscillation (NAO), typically resulting in winter warming across northern Eurasia. However, these changes remain highly uncertain, as they may coincide with strong internal variability in Northern Hemisphere wintertime climate. In contrast, Northern Hemisphere extratropical (NHET) eruptions are proposed to have opposite impacts, but they have been comparatively less studied, and large uncertainties remain regarding the ability of climate models to capture volcanic responses. This study examines winter North Atlantic climate responses to TROP and NHET eruptions by comparing temperature and atmospheric circulation patterns from last millennium simulations with multiple proxy-based reconstructions. 25 We find distinct differences in NAO-related climate changes in reconstructions, with TROP eruptions followed by a shift towards positive NAO and NHET eruptions associated with a negative NAO. In comparison, modelled responses exhibit a wide spread with strong dependence on the choice of volcanic forcing dataset. Notably, simulations using the latest volcanic forcing data show improved agreement with reconstructions, particularly for TROP eruptions. This model-proxy agreement 30 provides a useful basis for investigating the mechanisms that drive positive NAO responses after TROP eruptions. However, the simulated impacts of NHET eruptions are less consistent and remain unclear. These results highlight the importance of improved volcanic forcing datasets, refined paleoclimate reconstructions, and robust statistical approaches to better constrain uncertainties in assessing volcanic impacts on North Atlantic climate.

## 1 Introduction

55

60

Volcanic forcing is a major factor influencing natural climate variability on Earth (Fischer et al., 2007; Robock, 2000). Strong 35 volcanic eruptions inject a large amount of sulfur gases into the atmosphere and cause surface cooling by reducing incoming shortwave radiation, significantly perturbate the climate system and trigger severe societal impacts (Luterbacher and Pfister, 2015; Robock, 2000). Observations also indicate surface warming over Eurasia associated with a shift toward the positive phase of the North Atlantic Oscillation (NAO) during the subsequent 1-2 boreal winters following large tropical (TROP) eruptions (Christiansen, 2008; Robock and Mao, 1992). This post-eruption NAO shift is typically attributed to the enhanced 40 stratospheric meridional temperature gradient, linked to a strengthened stratospheric polar vortex and its downward propagation into the troposphere (Baldwin and Dunkerton, 1999; Driscoll et al., 2012; Swingedouw et al., 2017). The NAO, defined by the fluctuations in atmospheric pressure gradient between the Icelandic low and the Azores high, is a predominant mode of large-scale atmospheric circulation dominating the winter North Atlantic climate variability (Hurrell et al., 2003). Although the NAO variability primarily originates from internal atmospheric dynamics and ocean interaction (Rodwell et al., 45 1999), its observed positive shift following large TROP eruptions demonstrates its susceptibility to volcanic forcing, one of the strongest natural forcings of the climate system (Hurrell et al., 2003; Schneider et al., 2009). Therefore, understanding the NAO-associated climate responses after volcanic eruptions and evaluating the performance of climate models in simulating these changes are essential to distinguish how internal climate variability can be dynamically perturbed by natural forcing (Stenchikov et al., 2006), thereby improving confidence in future climate projections (Illing et al., 2018; Paik et al., 2023; 50 Swingedouw et al., 2017).

The positive NAO phase and associated winter Eurasian warming following TROP eruptions have been intensively studied through climate model simulations (Driscoll et al., 2012; Stenchikov et al., 2002, 2006; Zambri and Robock, 2016) and paleoclimate reconstructions (Fischer et al., 2007; Ortega et al., 2015; Shindell et al., 2004; Sjolte et al., 2018, 2021). However, large uncertainties persist due to the strong internal variability in Northern Hemisphere wintertime climate (Bittner et al., 2016; Swingedouw et al., 2017; Tejedor et al., 2024). While the Coupled Model Intercomparison Project Phase 5 (CMIP5) historical simulations demonstrate reasonable ability to reproduce the strengthening of the Northern Hemisphere polar vortex and winter Eurasian warming after the two largest TROP eruptions (the 1883 Krakatau and 1991 Pinatubo) since 1850 (Bittner et al., 2016; Zambri and Robock, 2016), such responses become less detectable when smaller eruptions are included (Driscoll et al., 2012). This raises critical questions regarding whether involving smaller eruptions may dilute the composited post-volcanic signals (Bittner et al., 2016; Zambri and Robock, 2016), or whether a large number of events is necessary to improve statistical

robustness and signal-to-noise ratio, thereby facilitating a clearer detection of externally forced responses amid strong internal climate variability (Shindell et al., 2004; Swingedouw et al., 2017).

In contrast to the extensive research on TROP eruptions, the climate impacts of Northern Hemisphere extratropical (NHET) eruptions have been comparatively less studied (Fuglestvedt et al., 2024; Paik et al., 2023). NHET eruptions may induce different climate impacts to TROP eruptions due to their distinct aerosol dispersion patterns and atmospheric residence times (Oman et al., 2005; Toohey et al., 2019; Zambri et al., 2019a). Simulations of the largest NHET event since 1850, the 1912 Katmai eruption, show a post-volcanic shift to negative NAO (Oman et al., 2005). Similar circulation changes have been identified in paleoclimate reconstructions (Sjolte et al., 2021) and in simulations of the 1783–1784 Laki eruption (Zambri et al., 2019a). Recent studies also suggest that NHET eruptions could exert a stronger impact on Northern Hemisphere temperature than TROP eruptions as their volcanic aerosols are more confined within one hemisphere (Toohey et al., 2019; Zhuo et al., 2021). Nevertheless, our understanding of the differences in climate responses to TROP and NHET eruptions remains limited, highlighting the need for further investigation.

The last millennium offers a more adequate record of volcanic eruptions to perform a more statistically robust analysis compared to the relatively short instrumental period (Fischer et al., 2007; Shen et al., 2025; Swingedouw et al., 2017; Zambri et al., 2017). Also, paleoclimate reconstructions can serve as benchmarks for cross-verifying model simulations (Ortega et al., 2015) in the Paleoclimate Modelling Intercomparison Project (PMIP) (Jungclaus et al., 2017; Schmidt et al., 2011). However, post-volcanic climate responses over the last millennium reveal considerable variability depending on the performance of individual models (Swingedouw et al., 2017; Zambri et al., 2017), the volcanic forcing data used by models (Zambri et al., 2017), the criteria for event selection (Wang et al., 2023), and confounding noises in paleoclimate reconstructions (Fischer et al., 2007; Ortega et al., 2015; Sjolte et al., 2021). Moreover, comparisons between simulated and reconstructed North Atlantic climate responses remain challenging, primarily due to the limited direct gridded reconstructions of atmospheric circulation (Fischer et al., 2007; Swingedouw et al., 2017), which may dilute or misattribute the atmospheric signals linked to volcanic forcing. Notably, recently developed gridded climate reconstructions that incorporate atmospheric pressure fields (Sjolte et al., 2018; Valler et al., 2021, 2024) offer the opportunity for detailed model-proxy comparisons to further resolve the volcanic impacts on atmospheric variability.

In this study, we use three of the latest gridded paleoclimate reconstructions covering the North Atlantic-European region (Sjolte et al., 2018; Tao et al., 2023; Valler et al., 2021, 2024) to evaluate winter North Atlantic climate responses to volcanic eruptions in 15 PMIP3 and PMIP4 last millennium simulations. The main focuses of this study are (1) to apply consistent criteria for event selection across different volcanic forcing datasets, (2) to assess the sensitivity of North Atlantic climate responses to eruption selections, and (3) to directly compare simulated responses against reconstructions to better characterize the different impacts between TROP and NHET eruptions on winter North Atlantic climate variability.

#### 2 Data and Methods





#### 2.1 Model data and paleoclimate reconstructions

We select 15 past1000 or past2k simulations (Table 1) to investigate the winter North Atlantic climate responses to volcanic eruptions over the last millennium (851-1850), including 10 models from the PMIP Phase 3 (PMIP3) and 5 models from the Phase 4 (PMIP4). The 15 selected PMIP simulations employed three volcanic forcing datasets with different spatial and temporal resolutions, the Gao-Robock-Ammann data (GRA, Gao et al., 2008), the Crowley data (CEA, Crowley and Unterman, 2013), and the eVolv2k data (Toohey & Sigl, 2017), with each forcing used by 5 models. The GRA data contain monthly sulfate aerosol loading from 9 km to 30 km at 0.5 km resolution for each 10° latitude band globally (Gao et al., 2008), while the CEA data provide stratospheric aerosol optical depth (AOD) at 550 nm with a time step of 10 days at four latitude bands: 30°S-90°S, 0°-30°S, 0°-30°N, 30°N-90°N (Crowley and Unterman, 2013). The latest eVolv2k data provide monthly stratospheric AOD at 96 latitudes globally, including estimated date, source latitude, and amount of stratospheric sulfur injection for each eruption (Toohey and Sigl, 2017). For the paleodata-model comparison, we use winter (December-February) sea level pressure (SLP) and near-surface air temperature (T2m) from our updated seasonally-resolved gridded reconstructions of the North Atlantic climate (SEA18v2, 1241–1970) (Sjolte et al., 2018; Tao et al., 2023). To further support data comparison, we also use two additional independent reconstructions, the monthly-resolved Modern Era Reanalysis (ModE-RA, 1421–2008) (Valler et al., 2024) and its previous version (EKF400v2, 1601–2005) (Valler et al., 2021). The SEA18v2 reconstructions are produced using an analogue-type method that matches the leading spatiotemporal variability of modeled isotopic composition to that measured in Greenland ice cores (Sjolte et al., 2018; Tao et al., 2023), whereas the ModE-RA and EKF400v2 datasets are produced using offline data assimilation approaches that integrate atmospheric model simulations with proxy records, documentary evidence, and instrumental measurements (Valler et al., 2021, 2024). However, these three reconstructions are not able to be used to examine the stratospheric-tropospheric structure as they only provide data at up to two vertical levels.

Table 1. PMIP3 and PMIP4 last millennium simulations analyzed in this study.

| Model name   | PMIP experiment | Atmospheric horizontal grids (Longtide Longitude × Latitude) | Volcanic forcing |
|--------------|-----------------|--------------------------------------------------------------|------------------|
| BCC-CSM1-1   | PMIP3, past1000 | 128×64                                                       | GRA              |
| CCSM4        | PMIP3, past1000 | 288×192                                                      | GRA              |
| FGOALS-s2    | PMIP3, past1000 | 128×108                                                      | GRA              |
| IPSL-CM5A-LR | PMIP3, past1000 | 96×96                                                        | GRA              |
| MRI-CGCM3    | PMIP3, past1000 | 320×160                                                      | GRA              |

| 120 | CSIRO-Mk3L-1-2 | PMIP3, past1000 | 64×56   | CEA     |
|-----|----------------|-----------------|---------|---------|
|     | EC-Earth3.1*   | PMIP3, past1000 | 320×160 | CEA     |
| 125 | HadCM3         | PMIP3, past1000 | 96×73   | CEA     |
|     | MIROC-ESM      | PMIP3, past1000 | 128×64  | CEA     |
|     | MPI-ESM-P      | PMIP3, past1000 | 192×96  | CEA     |
|     | ACCESS-ESM1-5  | PMIP4, past1000 | 192×145 | eVolv2k |
|     | INM-CM4-8      | PMIP4, past1000 | 180×120 | eVolv2k |
| 130 | MIROC-ES2L     | PMIP4, past1000 | 128×64  | eVolv2k |
|     | MRI-ESM2-0     | PMIP4, past1000 | 320×160 | eVolv2k |
|     | MPI-ESM1-2-LR  | PMIP4, past2k   | 320×160 | eVolv2k |

<sup>\*</sup> The EC-Earth3.1 past1000 simulation was completed after PMIP3 but still followed its protocol (Zhang et al., 2021).

# 2.2 Criteria of event selection for volcanic eruptions




Similar to previous model-proxy comparison work on volcanic impacts (Liu et al., 2022), we select eruptions in the PMIP last millennium simulations according to the volcanic aerosol datasets used to force the models, as three different datasets are employed across the 15 simulations. It is suggested for the eVolv2k data that eruptions with peak stratospheric aerosol optical depth over 30°N–90°N exceeding half of that of the 1991 Pinatubo eruption can potentially cause significant changes in the circum-North Atlantic climate (Wang et al., 2023). Therefore, here we adapt this criterion to select eruptions from the GRA, CEA, eVolv2k data separately for the analysis of last millennium simulations (Text S1), which are then categorized by the latitude of eruption to tropical (30°S–30°N, TROP) and Northern Hemisphere extratropical (>30°N, NHET) eruptions. In total, 18 TROP and 10 NHET eruptions in GRA data, 12 TROP and 19 NHET eruptions in CEA data, 21 TROP and 16 NHET eruptions in eVolv2k data during 851–1850 are selected (Table 2). For the proxy-based reconstructions, eruptions are identified from the volcanic forcing reconstruction of Sigl et al. (2015), which is derived from Greenland and Antarctic ice cores. To perform an independent analysis for paleo-reconstructions In total, 12 TROP eruptions with global volcanic aerosol forcing exceeding that of the 1991 Pinatubo eruption (-6.49 W m<sup>-2</sup>) and 8 NHET eruptions exceeding -1.8 W m<sup>-2</sup> (Text S2) during their full coverage period (1241–2008) are selected (Table 2). from the volcanic forcing reconstruction in Sigl et al. (2015).

Table 2. List of selected TROP and NHET eruptions. For the 15 PMIP last simulations, the eruptions during 851–1850 are selected from three volcanic forcing data, GRA (Gao et al., 2008), CEA (Crowley and Unterman, 2013), eVolv2k (Toohey and Sigl, 2017). For the three

paleoclimate reconstructions, eruptions are selected during their full coverage period (1241–2008) from the global volcanic aerosol forcing reconstructions in Sigl et al. (2015).

| Volcanic forcing   | TROP eruptions                     | NHET eruptions                     |  |
|--------------------|------------------------------------|------------------------------------|--|
|                    | 870, 901, 992, 1001, 1081, 1122,   | 933, 1176, 1328, 1459, 1584, 1719, |  |
| GRA                | 1167, 1195, 1227, 1258, 1284,      | 1729, 1755, 1783, 1831             |  |
|                    | 1452, 1600, 1641, 1761, 1809,      |                                    |  |
|                    | 1815, 1835                         |                                    |  |
| CEA                | 896, 971, 1257, 1286, 1330, 1441,  | 874, 911, 924, 1060, 1067, 1100,   |  |
|                    | 1600, 1673, 1809, 1815, 1831, 1835 | 1118, 1312, 1389, 1459, 1515,      |  |
|                    |                                    | 1525, 1553, 1585, 1667, 1731,      |  |
|                    |                                    | 1739, 1783, 1796                   |  |
|                    | 916, 976, 1028, 1108, 1171, 1191,  | 904, 939, 1020, 1182, 1200, 1210,  |  |
|                    | 1230, 1257, 1286, 1345, 1453,      | 1329, 1477, 1510, 1567, 1646,      |  |
| eVolv2k            | 1458, 1585, 1595, 1600, 1640,      | 1667, 1729, 1739, 1766, 1783       |  |
|                    | 1695, 1809, 1815, 1831, 1835       |                                    |  |
| Sigl et al. (2015) | 1258, 1276, 1286, 1345, 1458,      | 1329, 1477, 1646, 1667, 1729,      |  |
|                    | 1601, 1641, 1695, 1809, 1815,      | 1739, 1783, 1912                   |  |
|                    | 1836, 1991                         |                                    |  |
|                    |                                    |                                    |  |

# 2.3 Statistical analysis of post-volcanic climate responses



Empirical orthogonal function (EOF) analysis is used to calculate the winter (December–February, DJF) North Atlantic Oscillation (NAO) indices as the first principal components of winter SLP anomalies over the North Atlantic region (20°N–70°N, 90°W–40°E) (Hurrell et al., 2003). For the spatial patterns of post-volcanic SLP and T2m, composited anomalies are calculated relative to the five years preceding eruptions, and statistical significance at the 95% confidence level is assessed using a local two-tailed Student's t-test. In addition, superposed epoch analysis, a composite method to isolate signals from large background noise (Brad Adams et al., 2003; Driscoll et al., 2012; Wang et al., 2023; Zambri et al., 2017), is used to detect the post-volcanic responses of the NAO and regional T2m for each dataset. Similar to Liu et al. (2022), we identify the year of peak stratospheric sulfate aerosol loading as the key year for each eruption (Table S1) to ensure sufficient time for the volcanic aerosols to influence the subsequent winter climate. Here wWe use an 11-year time window with five years before and five years after the key year of each eruption and estimate statistical significance at the 95% confidence level by randomly

resampling 10,000 sets of pseudo-events from the data (Brad Adams et al., 2003). Year 0 in the superposed epoch analysis denotes the first DJF winter with January–February in the peak forcing year, which is defined as the first winter following the eruption.

#### **170 3 Results**



## 3.1 Post-volcanic changes in sea level pressure

Figure 1 shows the spatial patterns of winter SLP anomalies during the first winter following the large TROP and NHET eruptions listed in Table 2 which can potentially influence the North Atlantic region in three paleoclimate reconstructions as well as in three multi-model means of PMIP simulations driven by different volcanic forcing datasets. Overall, SLP anomalies during the first winter after TROP eruptions exhibit generally consistent features across reconstructions and PMIP simulations, displaying a north-south dipole, though minor spatial variations are evident (Fig. 1a–b). Specifically, SEA18v2 and ModE-RA reconstructions display clear positive NAO patterns (Fig. 1a), with pronounced negative SLP anomalies over Greenland and positive SLP anomalies spanning the mid-latitudes of the North Atlantic. This dipole pattern aligns well with the typical latitude (~55°N) that divides the two key regions of NAO (Stephenson et al., 2006). Although the EKF400v2 reconstruction exhibits a less distinct SLP anomaly pattern, it does show a weak tendency toward positive NAO (Fig. S1). Among the PMIP simulations (Fig. 1b), GRA-forced models reveal a positive NAO-associated SLP anomaly pattern after TROP eruptions. However, SLP anomalies in CEA-forced and eVolv2k-forced simulations more closely resemble the Scandinavian and the East Atlantic patterns, respectively (Comas-Bru and McDermott, 2014), rather than the typical NAO pattern.

In the first winter following NHET eruptions, SLP anomaly patterns exhibit greater variability among reconstructions and models compared to TROP eruptions (Fig. 1c–d). In SEA18v2 and EFK400v2 reconstructions (Fig. 1c), a north-south dipole with a positive center near Greenland and a negative center over the mid-latitudes emerges, indicating a negative NAO shift, while the ModE-RA reconstruction shows a slightly skewed dipole structure. In eVolv2k-forced simulations (Fig. 1d), a significant positive anomaly center appears over the subpolar North Atlantic, which resembles the negative NAO pattern and distinctly differs from the responses to TROP eruptions. In contrast, GRA-forced and CEA-forced simulations show no significant anomalies following NHET eruptions. Additionally, SLP anomalies in the second winter after eruptions are generally weaker and less consistent than those observed during the first winter, particularly in simulations (Fig. S2). In summary, the comparison of reconstructions and simulations provides clear evidence that winter SLP anomalies differ after TROP and NHET eruptions. Such differences highlight the need to analyze TROP and NHET eruptions separately to assess volcanic impacts on North Atlantic winter climate.

# (a) TROP, reconstructions

# (b) TROP, simulations

GRA (5)

# (c) NHET, reconstructions

SEA18v2

# (d) NHET, simulations

-2.0 -1.5 -1.0

SLP (hPa)

8

Figure 1. Spatial patterns of sea level pressure (SLP, units: hPa) anomalies in the first winter (December–February) following tropical (TROP) and Northern Hemisphere extratropical (NHET) volcanic eruptions. (a) SLP anomalies after TROP eruptions in three paleoclimate reconstructions (SEA18v2, ModE-RA, EKF400v2) and in (b) three multi-model ensemble means from PMIP simulations forced by GRA, CEA, and eVolv2k volcanic forcing datasets (numbers in brackets indicate the number of models forced by each dataset). (c–d) Same as (a–b), but for NHET eruptions. Anomalies are calculated relative to the mean conditions of the five years preceding the eruptions. Blue and red polygons (90°W–60°E, 55°N–90°N and 90°W–60°E, 20°N–55°N) are the two key regions of NAO (Stephenson et al., 2006). White dots indicate regions where anomalies are statistically significant at the 95% confidence level.

# 3.2 Post-volcanic changes in near-surface air temperature

Spatial patterns of T2m anomalies during the first winter after volcanic eruptions, particularly over northern Eurasia (Fig. 2), are closely linked to their corresponding SLP changes in Fig. 1. This relationship can be further supported by significant correlation coefficients (*R*>0.80) between the NAO indices and northern Eurasian T2m (55°N–70°N, 10°E–120°E) across all

reconstructions and simulations during their entire covering period. Significant Eurasian warming occurs during the first winter following TROP eruptions in SEA18v2 and ModE-RA reconstructions, also in GRA-forced models (Fig. 2a–b), associated with their positive NAO patterns (Fig. 1a–b). EKF400v2 reconstruction and eVolv2k-forced simulations display weaker and non-significant warming as their SLP anomalies slightly deviate from the canonical NAO pattern. Following NHET eruptions (Fig. 2c–d), a reversal in T2m patterns emerges, characterized by warming over Greenland and cooling across northern Eurasia.
This reversal is observed in reconstructions and eVolv2k-forced simulations, which also present SLP patterns that notably differ from those observed after TROP eruptions (Fig. 1). Overall, the tendencies toward Eurasian warming following TROP eruptions and Eurasian cooling following NHET eruptions are more evident and consistent in three reconstructions than PMIP simulations. These post-volcanic temperature changes are also clearly detectable in the reconstructions both through superposed epoch analysis to extract composite signals and by directly examining T2m anomalies after individual eruptions
(Fig. S3).

**Figure 2.** Same as Fig. 1, but for near-surface air temperature anomalies (T2m, unit: °C) in the first winter following TROP and NHET eruptions.

For the three multi-model ensemble means, the most pronounced temperature changes are the tropical cooling following TROP eruptions, as shown in their T2m time series over 0°–30°N (Fig. 3a). Compared to TROP eruptions, changes in tropical T2m after NHET eruptions are much weaker. Superposed epoch analyses further confirm that significant tropical cooling can persist for up to five winters following TROP eruptions, with peak anomalies exceeding -3°C (Fig. 3b), whereas cooling following NHET eruptions is substantially weaker, with peak anomalies less than half that magnitude (Fig. 3c). Notably, a linear relationship is identified between the magnitude of eruption and the amplitude of tropical cooling during the first post-volcanic

winter (Fig. 3d–f). Given that the standard deviations of tropical T2m changes in the multi-model means remain below 0.2°C, the tropical cooling driven by the shortwave radiation reductions due to volcanic eruptions therefore represents a noticeable perturbation to natural climate variability.

Conversely, post-volcanic winter temperature responses over northern Eurasia (55°N–70°N, 10°E–120°E) are ambiguous in simulations, complicating the identification of warming or cooling (Fig. 3g). This uncertainty stems from the higher background noise associated with greater natural variability and a wider inter-model spread compared to the tropical T2m. After applying superposed epoch analysis, a significant Eurasian winter warming of 0.32°C emerges in the first winter following TROP eruptions in GRA-forced models (Fig. 3h), consistent with the corresponding patterns of SLP and T2m anomalies (Fig. 1b, Fig. 2b). Specifically, the first-winter Eurasian warming after 8 out of 18 selected TROP eruptions exceeds the standard deviation (0.98°C) of the Eurasian T2m (Fig. 3j). Moreover, the Eurasian winter warming is not a simple linear atmospheric response (Fig. 3j–l) because it is linked to the downward propagation of polar vortex anomalies from the stratosphere to the troposphere (Baldwin and Dunkerton, 1999; Robock, 2000; Robock and Mao, 1992). Internal variability of the North Atlantic climate can further obscure or suppress such signal (Swingedouw et al., 2017). However, the changes in Eurasian T2m following NHET eruptions remain inconclusive across multi-models means (Fig. 3i–l).

240

**Figure 3.** Winter near-surface air temperature responses to TROP and NHET eruptions in the last millennium simulations. (a) Time series of winter near-surface air temperature (T2m, unit: °C) anomalies over the tropical region (0°–30°N) for multi-model means forced by GRA, CEA, eVolv2k datasets. Grey shading denotes the spread among individual model simulations. Red and blue triangles indicate TROP and NHET eruptions with magnitudes greater than the 1991 Pinatubo eruption defined in the eVolv2k dataset, respectively. (b-c) Results of superposed epoch analysis for tropical T2m over 0°–30°N after TROP and NHET eruptions, with data statistically significant at the 95% confidence level marked with asterisk. (d-f) T2m anomalies over 0°–30°N in the first winter following each eruption with respect to the mean of the last millenniumaverage of five years before the event. The solid lines represent the linear regression of T2m anomalies against the magnitude of eruptions, with star symbols indicating that linear trends are statistically significant at the 95% confidence level using F-test. (g-l) are as (a-f) but for the regional T2m over northern Eurasia (55°N–70°N, 10°E–120°E).

The multi-model means from the PMIP simulations forced by GRA, CEA, eVolv2k datasets exhibit different SLP and T2m responses to TROP and NHET eruptions (Fig. 1–3), indicating that simulated post-volcanic North Atlantic climate responses in climate models are dependent on the choice of volcanic forcing data. The monthly evolution of volcanic forcing over the extratropical Northern Hemisphere (30°N–90°N) for the selected TROP eruptions reveals distinct patterns among the datasets in terms of the timing of peak forcing and the duration of peak (Fig. 4a). In the GRA dataset, most TROP events exhibit a linear buildup and reach peak forcing approximately 4 months after eruption. In contrast, the CEA dataset shows a more delayed peak occurring around 8 months after eruption. The eVolv2k dataset displays a different pattern, where most events

reach a weak plateau around 4 months, followed by a peak approximately 12 months after eruption. For NHET eruptions, peak forcing in the eVolv2k data also occurs later than in the GRA and CEA datasets (Fig. 4b). These differences in volcanic forcing datasets can introduce additional uncertainties in the analysis of post-volcanic climate responses, especially when focus on a specific season, for example, the responses of winter NAO during the 1-2 years following eruptions. Another notable difference among the forcing datasets lies in their spatial resolution, as shown by the spatiotemporal distribution of global volcanic forcing for the largest TROP eruption (the 1257 Samalas eruption) and the largest NHET eruption (the 1783 Laki eruption) over the last millennium. These distributions reveal differences in both the latitudes of eruption source and peak forcing (Fig. 4c-d). The Samalas eruption is identified as occurring in 1258 in the GRA dataset, where the mid-latitude aerosol forcing becomes significantly large about half a year before the following 1258/1259 DJF winter. In contrast, both the CEA and eVolv2k datasets assign the onset of the eruption to 1257. In particular, volcanic aerosol forcing in the CEA dataset begins only about two months before the 1257/1258 DJF winter, with its maximum forcing occurring afterward. This timing suggests that the aerosols may not have sufficient time to influence the climate during the first winter following the eruption, which could further complicate the comparisons among models forced by different datasets. Additionally, the 1783 Laki eruption appears less pronounced in the CEA dataset compared to the GRA and eVolv2k datasets (Fig. 4b, d). This discrepancy arises because the CEA data consider most of Laki's forcing to originate in the troposphere rather than stratosphere (Crowley and Unterman, 2013). We further compare post-volcanic temperature changes across individual models and find distinct patterns among models forced by the same data, indicating that climate responses to volcanic eruptions are also model-dependent (Fig. 5). Consistent with the multi-model mean responses (Fig. 2-3), the most robust and coherent signal across models is tropical cooling after TROP eruptions. Also, the models tend to overestimate the tropical cooling compared with the anomalies after the 1991 Pinatubo eruption in the 20th Century Reanalysis version 3 (20CRv3, Slivinski et al., 2019), a bias also reported in CMIP5 historical simulations (Driscoll et al., 2012). In contrast, temperature responses over Eurasia are more variable, even among models using the same volcanic forcing. Overall, three GRA-forced models (BCC-CSM1-1, FGOALS-s2, MRI-CGCM3) tend to exhibit tropical cooling and Eurasian winter warming following TROP eruptions (Fig. 5), while no clear agreement is found among the CEA-forced and eVolv2k-forced models. Only the eVolv2k-forced MPI-ESM1-2-LR model showing distinct Eurasian warming following TROP eruptions and cooling following NHET eruptions, consistent with reconstructions.





Figure 4. Evolution of monthly volcanic forcing over the extratropical Northern Hemisphere (30°N–90°N, with latitude-weighted) for the selected (a) TROP and (b) NHET eruptions from GRA, CEA, and eVolv2k datasets listed in Table 2. The strength of volcanic forcing is represented by total volcanic aerosol loading from 9 km to 30 km in GRA (unit: kg sulfate aerosol/km²), and by stratospheric aerosol optical depth (AOD) at 550 nm in CEA and eVolv2k. Triangles indicate the month of peak forcing following each eruption. Thicker lines denote the largest TROP eruption (the 1257 Samalas eruption) and the largest NHET eruption (the 1783 Laki eruption) of the last millennium. Panels (c) and (d) show the spatial and temporal distribution of global volcanic forcing in three datasets for the Samalas eruption and Laki eruption, respectively.

Figure 5. Near surface air temperature anomalies in the first winter after each cruption in PMIP last millennium simulations. Each subplot shows the near surface air temperature (T2m, unit: °C) anomalies over 0° 30°N and northern Eurasia (55°N 70°N, 10°E 120°E) in the first winter after each TROP and NHET cruptions with respect to the average of five years before the event. The gray dashed lines denote the ±σ and ±2σ of T2m changes over northern Eurasia in the last millennium. The sizes of the circles represent the magnitudes of the cruptions which are normalized to the Samalas cruption.

Figure 5. Near-surface air temperature (T2m, unit: °C) anomalies during the first winter after each eruption in PMIP last millennium simulations. Each subplot shows the T2m anomalies over 0°-30°N and northern Eurasia (55°N-70°N, 10°E-120°E) relative to the mean of

# 3.3 Sensitivity of post-volcanic signals to the eruption event selection








Here we systematically examine the sensitivity of post-volcanic NAO and associated northern Eurasian temperature responses to eruption event selection in both reconstructions and PMIP last millennium simulations. Similar to the reconstructions (Fig. S1), the PMIP simulations also require a sufficient number of eruption events to derive statistically robust changes in the NAO and corresponding Eurasian temperature anomalies during the first post-volcanic winter (Fig. 6). After TROP eruptions, significant positive NAO and Eurasian warming consistently emerge in two GRA-forced models, BCC-CSM1-1 and MRI-CGCM3, when including at least 1211 and 8 of the largest events, respectively (Fig. 6a-b). Although the FGOALS-s2 model also demonstrates persistent positive NAO anomalies, its associated Eurasian warming signal notably weakens with the inclusion of additional smaller eruptions. Conversely, the five CEA-forced models fail to produce significant NAO or Eurasian temperature changes, consistent with results from multi-model means (Fig. 1-3) and individual eruptions (Fig. 5). A general but non-significant tendency toward the positive NAO is found in the eVolv2k-forced models except for INM-CM4-8, and MPI-ESM1-2-LR exhibits a particularly distinct and significant Eurasian warming. In contrast, responses after NHET eruptions exhibit greater divergence across models compared to TROP eruptions (Fig. 6c-d). Three eVolv2k-forced models (ACCESS-ESM1-5, MRI-ESM2-0, MPI-ESM1-2-LR) show a tendency toward negative NAO associated with Eurasian cooling, although not all signals are significant. These results agree with a previous evaluation of PMIP3 models, which found that the simulated responses to eruptions strongly depend on both the individual models and the choice of volcanic forcing data (Zambri et al., 2017).

To further evaluate the robustness of post-volcanic signals and avoid potential dominance by large events, the NAO and Eurasian T2m anomalies are then analyzed by incrementally adding eruptions in order of increasing forcing magnitude (Fig. 6e–h). Results reveal that NAO and Eurasian temperature responses depend on eruption magnitude, with some models exhibiting reversed anomaly signs when eruptions are added in the opposite order. Notably, two GRA-forced models (BCC-CSM1-1 and MRI-CGCM3) consistently maintain their significant positive NAO anomalies and Eurasian warming in response to TROP eruptions (Fig. 6e–f). Furthermore, the eVolv2k-forced MPI-ESM1-2-LR model exhibits stable signals of Eurasian warming after TROP eruptions (Fig. 6f) and cooling after NHET eruptions (Fig. 6h). Their corresponding spatial patterns of SLP and T2m in the first post-volcanic winter are all consistent with these shifts (Fig. S4). Based on both the analysis of individual eruptions (Fig. 5) and superposed epoch analysis with different event selections (Fig. 6), the stable post-volcanic signals in these three models can be interpreted as genuine responses to volcanic forcing rather than statistical artifacts. Furthermore, the second post-volcanic winter exhibits a dominance of widespread cooling (Fig. S5), in line with the CMIP5

historical simulations that winter Eurasian warming is typically limited to the first winter following TROP eruptions (Zambri and Robock, 2016).

**Figure 6.** Responses of North Atlantic climate in the first winter to TROP and NHET eruptions in PMIP last millennium simulations. Results from superposed epoch analysis of (a) NAO and (b) T2m over northern Eurasia (55°N–70°N, 10°E–120°E) after TROP eruptions, with the order by adding eruptions from large to small peak forcing. (c-d) are the same as (a-b) but for NAO and T2m after NHET eruptions. (e-h) are the same as (a-d) but with the order by adding eruptions from small to large peak forcing. Data statistically significant at the 95% confidence level are marked with white dots.

#### 3.4 Changes in atmospheric circulation after TROP and NHET eruptions

375

380

The three models that exhibit relatively stable NAO and Eurasian T2m responses (BCC-CSM1-1, MRI-CGCM3, MPI-ESM1-2-LR) are further examined to investigate the changes in stratospheric-tropospheric circulation (Fig. 7). After TROP eruptions, all three models show distinct strengthened westerlies extending from the stratosphere into the troposphere, along with negative zonal wind anomalies around 30°N–40°N (Fig. 7a). These patterns align with the proposed dynamical response involving a strengthened stratospheric polar vortex that propagates downward and induces a positive NAO signal in SLP (Baldwin and Dunkerton, 1999; Swingedouw et al., 2017). In addition, the proposed temperature changes induced by TROP eruptions are also clearly captured by these models (Fig. 7b), with tropical surface cooling due to the reduced incoming shortwave radiation and warming in the lower stratosphere from the increased absorption of solar near-infrared radiation and upwelling longwave radiation (Driscoll et al., 2012; Robock, 2000; Swingedouw et al., 2017). Consistent with a previous evaluation of GRA-forced and CEA-forced PMIP3 simulations (Zambri et al., 2017), the latest eVolv2k-forced PMIP4 models without significant NAO and Eurasian T2m responses on the surface do still show a strengthened polar vortex in the stratosphere but fail to propagate this change downward (Fig. S6).

After NHET eruptions, the MPI-ESM1-2-LR model shows the most consistent response with a significant negative NAO shift associated with Eurasian cooling (Fig. 6), aligning well with paleoclimate reconstructions (Fig. 1–2) and simulations of the 1912 Katmai eruption at 58°N (Oman et al., 2005). Its simulated atmospheric circulation changes exhibit a pattern of positive-negative-positive zonal winds anomalies from tropics through mid-latitudes and to polar regions in the troposphere (Fig. 7c), with a northward shift compared to the patterns after TROP eruptions (Fig. 7a). Such circulation responses are similar to those in the simulation of 1783–1784 Laki eruption (Zambri et al., 2019a), indicating an equatorward movement of the subtropical jet stream. The temperature anomalies in MPI-ESM1-2-LR after NHET eruptions also resemble those in the Laki simulation (Zambri et al., 2019a), with stratospheric heating and tropospheric cooling concentrated around 30°N–40°N (Fig. 7d). In contrast, BCC-CSM1-1 displays weak and ambiguous zonal wind and temperature anomalies after NHET eruptions, and MRI-CGCM3 continues to exhibit the patterns similar to those observed after TROP eruptions (Fig. 7c–d).

**Figure 7.** Northern Hemispheric zonal mean anomalies of (a) zonal wind (U, unit: m/s) and (b) air temperature (T, unit: °C) from the stratosphere to the troposphere in three selected PMIP models with winter Eurasian warming responses during the first winter following TROP eruptions. (c-d) are the same as (a-b) but for NHET eruptions. Anomalies are computed relative to the average of five years before the eruptions. White dots indicate regions where anomalies are statistically significant at the 95% confidence level.

Overall, the MPI-ESM1-2-LR model demonstrates skill in capturing volcanic impacts on the winter North Atlantic climate in its last millennium simulation with the PMIP4 volcanic forcing, particularly after TROP eruptions. To further assess its performance, we analyse 50 ensemble members from its historical simulation (1850–2014) with CMIP6 forcing to examine post-volcanic SLP and T2m changes. This evaluation focuses on the two largest TROP eruptions since 1850 (the 1991 Pinatubo and 1883 Krakatau), as previous studies have shown that CMIP5 models can reproduce a strengthened Northern Hemisphere

polar vortex and associated winter warming when only considering these two events (Bittner et al., 2016; Zambri and Robock, 2016). The ensemble means of the MPI-ESM1-2-LR historical runs exhibit a clear positive NAO pattern and Eurasian warming after both eruptions (Fig. 8). Nearly all members simulate post-volcanic tropical cooling signal. However, due to the strong natural variability of the Northern Hemisphere winter climate, as previously shown in Figure 3, the T2m anomalies over northern Eurasia display a large spread across the 50 ensemble members. Despite this wide spread, more members show Eurasian warming than cooling, and 17 members after Pinatubo and 13 members after Krakatau exhibit a warming signal that exceeds one standard deviation of their temperature variability. These results further indicate that, although a large spread exists, the MPI-ESM1-2-LR model performs well in reproducing post-volcanic wintertime climate responses.

Figure 8. (a) Sea level pressure (SLP) and (b) near-surface air temperature (T2m) anomalies in the 50-member ensemble mean of the MPI-ESM1-2-LR historical simulation after the 1991 Pinatubo eruption. White dots indicate regions where more than 30 of 50 members agree on the sign of the anomalies. (c) T2m anomalies over 0°–30°N and northern Eurasia (55°N–70°N, 10°E–120°E) relative to the mean of the historical simulation period (1850–2014), with red circles denoting individual members and the blue diamond denoting the ensemble mean. T2m anomalies over northern Eurasia exceeding one standard deviation (σ) are filled with color. Panels (d–f) are the same as (a–c) but for the 1883 Krakatau eruption.

#### 4 Discussion

# 4.1 Comparison of post-volcanic changes in PMIP simulations and reconstructions

Our study expands on previous work (e.g., Fischer et al., 2007; Swingedouw et al., 2017; Zambri et al., 2017) by directly comparing the spatial patterns of post-volcanic anomalies between climate models and reconstructions, providing a more detailed evaluation of winter North Atlantic climate responses to volcanic eruptions over the last millennium. By employing multiple statistical approaches for cross-validation, we also systematically assess the robustness of post-volcanic climate signals and their sensitivity to volcanic event selections. Overall, the three reconstructions analyzed here exhibit broadly consistent responses to volcanic eruptions, characterized by positive NAO-associated SLP and T2m patterns following TROP eruptions, and negative NAO-associated patterns after NHET eruptions (Fig. 1–2). Some structural differences are observed, which can be attributed to the different methodologies and datasets used to generate each reconstruction. Particularly, SEA18v2 is primarily based on the analogue matching of isotopic composition of Greenland ice cores and an isotope-enabled model simulation (Sjolte et al., 2018; Tao et al., 2023), while ModE-RA and EKF400v2 are paleoclimate data assimilation products that integrate not only proxy records but also documentary evidence and instrumental records (Valler et al., 2021, 2024). The presence of common post-volcanic climate signals across these reconstructions provide evidence for the last millennium that TROP and NHET eruptions have distinct influences on the winter North Atlantic climate. These reconstruction-based signals offer valuable references for paleodata-model comparison and model evaluation.

425

420

Among the 15 PMIP3-PMIP4 last millennium simulations examined here, we find that the models demonstrate both capabilities and limitations in capturing the North Atlantic climate responses to volcanic eruptions. Although discrepancies exist and not all features are fully resolved, several models generally simulate post-volcanic changes that align with paleoclimate reconstructions and capture the proposed dynamical processes in the stratosphere and troposphere. The agreement between some simulations and reconstructions provides a useful basis for investigating the potential mechanisms that may drive the distinct NAO responses to TROP and NHET eruptions. It is also noteworthy that the post-volcanic anomalies in the North Atlantic climate tend to persist for a shorter duration in the simulations compared to reconstructions (Fig. S1), consistent with previous studies (Ortega et al., 2015; Sjolte et al., 2021). This suggests that current climate models may have potential limitations in representing necessary atmosphere-ocean feedbacks required to sustain those atmospheric anomalies (Schneider et al., 2009).

# 4.2 Potential mechanisms of different NAO responses to TROP and NHET eruptions

The mechanisms behind the positive NAO and associated Eurasian winter warming following TROP eruptions have been extensively studied: warming in the tropical lower stratosphere enhances the equator-to-pole temperature gradient and

strengthens stratospheric westerlies according to the thermal wind relationship, leading to a strengthened stratospheric polar vortex that propagates downward and produces a positive NAO signal in SLP (e.g, Driscoll et al., 2012; Swingedouw et al., 2017, and reference therein). Our results demonstrate that the PMIP simulations with positive NAO and Eurasian winter warming capture these dynamical processes well (Fig. 7a\_b, e), supporting the proposed mechanism of how tropical eruptions influence North Atlantic winter climate variability. In contrast, NHET eruptions differ from TROP eruptions primarily in the latitude of SO<sub>2</sub> injection, shifting the center of volcanic aerosol-induced stratospheric warming from the tropics to the extratropical areas. These differences in thermal responses are clearly represented in the MPI-ESM1-2-LR model (Fig. 7b, d). As a result, NHET eruptions may in principle weaken the Northern Hemisphere stratospheric polar vortex and induce a negative NAO, as seen in simulations of the 1912 Katmai eruption (Oman et al., 2005). However, in simulations of the 1783 Laki eruption, the most significant circulation changes occur in the troposphere near 30°N and 200 hPa rather than in the stratosphere, with an equatorward shift of the subtropical jet that relates to a negative NAO (Zambri et al., 2019a) and closely resembles the pattern in the MPI-ESM1-2-LR model (Fig. 7c-d).

In addition to the negative NAO, a positive phase of El Niño-Southern Oscillation (ENSO) has also been identified in the Laki eruption simulations (Zambri et al., 2019b). This El Niño-like response, driven by a southward shift of the Intertropical Convergence Zone, has been observed in both model simulations and proxy records following tropical and Northern Hemisphere eruptions (Liu et al., 2018; Pausata et al., 2015b; Ward et al., 2021). Compared to tropical eruptions, Northern Hemisphere eruptions tend to produce stronger El Niño-like sea surface temperature anomalies and enhanced tropical convection (Liu et al., 2018). The intensification of ENSO-related tropical convection could support the development of a negative NAO during wintertime by reinforcing extratropical atmospheric Rossby wave trains (Geng et al., 2024), potentially contributing to the contrasting NAO responses seen after TROP and NHET eruptions. However, the NAO responses to volcanic eruptions are highly model-dependent, and in our results only the MPI-ESM1-2-LR model captures the negative NAO after NHET eruptions in agreement with reconstructions, highlighting substantial uncertainty in the mechanisms linking NAO variability to high-latitude eruptions. Also, previous studies suggest that a large ensemble size is needed to robustly detect post-volcanic climate changes against the strong winter climate variability in models (Bittner et al., 2016; Oman et al., 2005; Pausata et al., 2015a), which is rather challenging for the evaluation of PMIP simulations here typically only a single ensemble member is available for each model.

# 4.3 Forcing- and model-dependent volcanic imprints on winter NAO





Our results of PMIP3-PMIP4 models corroborate the earlier evaluation of PMIP3 models that simulated climate responses to volcanic eruptions are strongly dependent on the choice of volcanic forcing dataset (Zambri et al., 2017). Despite consistent criterion is applied in our selection of eruptions for analysis, discrepancies remain in the number of selected events and the timing of individual eruptions (Table 2), posing inherent challenges for model comparison when different volcanic forcing datasets are used (Zambri et al., 2017; Zhuo et al., 2014). The temporal evolution and spatial distribution of volcanic forcing

of the selected eruptions also differ markedly among the GRA, CEA, and eVolv2k datasets (Fig. 4), and these differences are critical because the climatic impact of an eruption depends on when and where its stratospheric aerosols evolve. For many eruptions without a recorded eruption month, the events are defined as occurring in April in the GRA dataset, but in January in the CEA and eVolv2k datasets. Such discrepancies in the timing and duration of peak forcing complicate the comparison of winter signals, as it becomes difficult to ensure both sufficient time for the eruptions to exert their impacts and an accurate capture of the peak climate anomalies. This challenge is particularly pronounced for NHET eruptions, which are often less well documented and identified than TROP eruptions due to their smaller magnitudes and more geographically restricted impacts on the Northern Hemisphere. As a result, the simulated impacts of NHET eruptions are less consistent across models compared to those of TROP eruptions. Overall, the GRA-forced models exhibit a more coherent NAO response to TROP eruptions than the CEA-forced and eVolv2k-forced models (Fig. 6), likely due to the larger amplitude of major TROP eruptions in GRA dataset (Toohey & Sigl, 2017). This can be supported by the comparison of the the GISS-E2-R simulations with CEA and 2×GRA forcing, where only the latter reproduces significant climate responses to TROP eruptions (Zambri et al., 2017). Notably, the latest PMIP4 simulations using the eVolv2k dataset, particularly MPI-ESM1-2-LR, show improved agreement with paleoclimate reconstructions in their responses to both TROP and NHET eruptions (Fig. 1-2, Fig. 6). This improved ability to distinguish the climate impacts of TROP and NHET eruptions may stem from the higher spatial resolution and more refined eruption identification of eVolv2k data, as the polar vortex response is sensitive to the structure of volcanic forcing (Toohey et al., 2014). Indeed, the same model run with different volcanic aerosol forcings for the same eruption can yield different patterns of circulation anomalies (Toohey et al., 2014). It has also been suggested that current volcanic forcing reconstructions of NHET eruptions are biased (Fuglestvedt et al., 2024, 2025). These limitations emphasize the need for more accurate volcanic forcing reconstructions to be consistently used by models (Fuglestvedt et al., 2024; Marshall et al., 2022), which could facilitate more consistent inter-model comparisons for assessing volcanic impacts.







The climate responses to volcanic eruptions are also strongly model-dependent. Even when using the same volcanic forcing dataset, models can exhibit substantial discrepancies, sometimes even with opposite post-volcanic North Atlantic climate responses (Fig. 6), which are often attributable to the differences in the representation of physical processes and aerosol treatments (Zanchettin et al., 2016). For example, the MRI-CGCM3 model produces the strongest responses following TROP eruptions among the GRA-forced simulations, likely due to its interactive stratospheric aerosol conversion scheme (Yukimoto et al., 2012), which is important for climate models to better capture the changes in climate variability (Pausata et al., 2015a). Additionally, while some models simulate stratospheric warming and polar vortex strengthening after TROP eruptions, they differ in both the magnitude of the anomalies and the vertical structure of the atmospheric response, indicating diverse pathways of stratosphere-to-troposphere coupling (Fig. 7). These differences further contribute to spatial inconsistencies in the representation of the north-south dipole in SLP anomalies (Fig. S4). For example, MPI-ESM1-2-LR reasonably captures the stratospheric-tropospheric signals and shows stable Eurasian winter warming, but its SLP pattern more closely resembles the East Atlantic pattern rather than the NAO. Such discrepancies are crucial not only for the downward propagation of the polar

vortex anomalies (Baldwin and Dunkerton, 1999) but also for contributing to the inter-model spread in NAO variability (Bonnet et al., 2024). Model biases in the spatial variability of leading North Atlantic climate modes (Tao et al., 2023) may be linked to the patchy simulated patterns of post-volcanic SLP anomalies. The reduction in wind speed following volcanic eruptions, attributed to weakened vertical momentum transport, may also interact with atmospheric circulation changes including the NAO (Shen et al., 2025).

## 4.4 Effects of the selection of eruptions for the detection of post-volcanic climate responses

It is particularly important to emphasize that the detection of post-volcanic climate responses is highly sensitive to the selection of volcanic eruptions for analysis (Bittner et al., 2016; Wang et al., 2023). Our results indicate no consensus on the optimal sample size to best capture the climate signals in either reconstructions or simulations (Fig. S1, Fig. 6). For example, while a previous study found no significant response in BCC-CSM1-1 following the 10 largest TROP eruptions over the last millennium (Zambri et al., 2017), our analysis reveals a robust positive NAO shift and associated Eurasian warming when considering 1211 or more events (Fig. 6a–b). The post-volcanic signals derived from superposed epoch analysis also have been questioned, particularly the Eurasian warming after TROP eruptions, arguing that climate conditions after individual eruptions should be inspected separately against natural climate variability rather than compositing signals from multiple events (Tejedor et al., 2024). These all underscores the uncertainties involved in applying statistical approaches to detect volcanic impacts (Marshall et al., 2022) and highlights the importance of testing different criteria for event selection (Bittner et al., 2016; Wang et al., 2023). Our analyses demonstrate that multiple statistical approaches, such as composite analysis, superposed epoch analysis, and event-by-event inspection, can be combined to cross-validate the robustness of post-volcanic signals. It is also important to examine the stratospheric-tropospheric circulation to distinguish genuine dynamical responses from potential statistical artifacts that might arise from sampling coincidences.

# **5 Summary**







Our study presents a detailed paleoclimate data-model comparison that provides new evidence of distinct North Atlantic climate responses following TROP versus NHET eruptions. By integrating PMIP3-PMIP4 last millennium simulations with three climate field reconstructions, we systematically evaluate the spatial and temporal structure of post-volcanic anomalies, with a particular focus on winter NAO-related changes. The combined use of multiple reconstructions and statistical approaches to cross-validate post-volcanic climate signals improves the robustness of the detected responses and provides a benchmark for model evaluation. The PMIP simulations show that, although models can capture important dynamical possesses following volcanic eruptions, the resulting climate responses depend strongly on the choice of volcanic forcing dataset, model configuration, and the selection of eruption events. However, the PMIP last millennium experiments do not offer output from simulations with varying volcanic forcing datasets or stratospheric aerosol schemes within the same model, which limits the ability to isolate the drivers of model response differences. Other dedicated model intercomparison projects,

such as the Volcanic Forcing Model Intercomparison Project (VolMIP, Zanchettin et al., 2016, 2022) and the Interactive Stratospheric Aerosol Model Intercomparison Project (ISA-MIP, Timmreck et al., 2018), offer more targeted frameworks for a technical diagnose of model discrepancies. To address remaining uncertainties in the assessment of climate impacts of volcanic eruptions, future work needs to prioritize developing improved volcanic forcing datasets, refining climate model representations of critical physical processes, and enhancing the quality and coverage of proxy-based reconstructions for robust model evaluation.

#### Data availability


The PMIP3 and PMIP4 simulations are available from the Earth System Grid Federation (ESGF) at https://esgf-node.llnl.gov/projects/cmip5/ and https://esgf-node.llnl.gov/projects/cmip6/. Our updated gridded climate reconstructions SEA18v2 (Sjolte et al., 2018; Tao et al., 2023)(Sjolte et al., 2018; Tao et al., 2023) are archived from Zenodo at https://zenodo.org/records/8328301. The monthly paleo-reanalysis data ModE-RA (Valler et al., 2024) and EKF400v2 (Valler et al., 2021) are available from the World Data Center for Climate (WDCC) at https://www.wdc-climate.de/ui/entry?acronym=ModE-RA s14203-18501 and https://www.wdc-climate.de/ui/entry?acronym=EKF400 v2.0.

# 555 Author contribution

QT and JS conceptualized the study. QT performed the analysis and wrote the original draft of the manuscript. CS contributed to the writing. RM and JS provided supervision. All authors reviewed and revised the manuscript.

# **Competing interests**

The authors declare that they have no conflict of interest.

### 560 Acknowledgments


This work was supported by the Swedish Energy Agency (Grant No. 51375-1) and the strategic research program of ModElling the Regional and Global Earth system (MERGE) hosted by the Faculty of Science at Lund University. Cheng Shen is also supported by the Formas (2023-01648), Stiftelsen Längmanska kulturfonden (BA24-0484), Stiftelsen Åforsk (24-707), and Adlerbertska Forskningsstiftelsen (AF2024-0069). We thank the climate modelling groups listed in Table S1 for producing and providing access to their PMIP simulations. We also thank Prof. Qiong Zhang from Stockholm University for sharing the past1000 simulations of EC-Earth 3.1.

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
