# Peer review of "Distinct winter North Atlantic climate responses to tropical and extratropical eruptions over the last millennium in PMIP simulations and reconstructions"

_EGUsphere, 2025_

## Author Response (AR1)

Comments of reviewers are provided in *italic blue*, our corresponding replies are in black, and the revised or newly added text in the manuscript is highlighted in red.

**Reply to reviewer 1:**

*Summary: The study analyses the response of volcanic eruptions in a set of climate model simulations, comparing them with proxy-based temperature and circulation reconstructions, over the past millennium. The study distinguishes between tropical and extratropical eruptions in the Northern Hemisphere. The main results are, in my interpretation, that the qualitative response is, first, strongly dependent on whether or not the eruptions have been tropical; it also strongly depends on the model, and on the forcing data set of past volcanic aerosols.*

*The study is motivated by a series of prior publications that also target the response of the atmospheric circulation to volcanic eruptions, which have found somewhat contradictory results. Whereas it seemed clear that the reconstructions did show a shift towards a positive NAO state and thus an Eurasian warming after eruptions. The modelled response was not that clear. Some studies, e.g., Tejedor et al. and Polvani et al., suggested that there was no response at all, and that the signal from the reconstructions, at least at the surface level, is just random internal variability.*

*This study offers an alternative interpretation, suggesting that the unclear signal derived from simulations may be due to differences among models and to differences in the volcanic aerosol data sets used to force the models. They find that the more recent forcing data, along with a focus on the most realistic models, allow them to identify a consistent signal, particularly for tropical eruptions.*

*Recommendation: I found the study interesting, and the manuscript is well-written. It can be somewhat controversial, as it does not agree with those previous studies, but I am happy to recommend it for publication. I have a few suggestions that the authors may want to consider:*

**Response:**

Thank you for the positive evaluation regarding our manuscript. Below, we provide our point-by-point responses.

*1) If my interpretation of the framing of the study is correct, I would suggest stating more clearly in the abstract the conceptual links to those prior analyses, explaining briefly why this study is important. The current abstract sounds correct, but it does not clearly convey the present backdrop and where this study fits.*

**Response:**

Thank you for bringing it up. We agree that the abstract can be improved. We revise the abstract it as follows:

**Abstract.** Large tropical (TROP) volcanic eruptions can influence North Atlantic climate by inducing a positive shift of the North Atlantic Oscillation (NAO), typically resulting in winter

warming across northern Eurasia. However, these changes remain highly uncertain, as they may coincide with strong internal variability in Northern Hemisphere wintertime climate. In contrast, Northern Hemisphere extratropical (NHET) eruptions are proposed to have opposite impacts, but they have been comparatively less studied, and large uncertainties remain regarding the ability of climate models to capture volcanic responses. This study examines winter North Atlantic climate responses to TROP and NHET eruptions by comparing temperature and atmospheric circulation patterns from last millennium simulations with multiple proxy-based reconstructions. We find distinct differences in NAO-related climate changes in reconstructions, with TROP eruptions followed by a shift towards positive NAO and NHET eruptions associated with a negative NAO. In comparison, modelled responses exhibit a wide spread with strong dependence on the choice of volcanic forcing dataset. Notably, simulations using the latest volcanic forcing data show improved agreement with reconstructions, particularly for TROP eruptions. This model-proxy agreement provides a useful basis for investigating the mechanisms that drive positive NAO responses after TROP eruptions. However, the simulated impacts of NHET eruptions are less consistent and remain unclear. These results highlight the importance of improved volcanic forcing datasets, refined paleoclimate reconstructions, and robust statistical approaches to better constrain uncertainties in assessing volcanic impacts on North Atlantic climate.

*2) In my opinion, one key figure of the manuscript is Figure 5 (also Figure 6). However, I had to stare at Figure 5 for a long time to fully grasp the message. First, it won't be easy for many readers to see all the details. The purportedly grey dashed lines showing the sigma and 2xsigma bounds can barely be discerned; actually, what the reader sees are other grey dashed lines marking the zero anomalies for both axes. Additionally, the zero grey lines are not visible in all panels, and the reader may wonder if there is a hidden meaning behind this omission. The explanation of the circle colour is also left to the figure inlet in the first panel, and it is not mentioned in the caption. The caption also states that the circle sizes, displaying the magnitude of the eruptions, are normalised to the Samalas eruption. Thus each panel should have one largest red circle of the same size (the Samalas eruption). However, this does not seem to be the case for all panels, e.g., not for MPI-ESM-P CEA or ACCES evolv2k.*

*My recommendation is that the authors give further thought to this complex yet important figure.*

**Response:**

Thank you for the advice on Figure 5. In the revised version, we change the patterns for the zero anomalies on both axes, as well as the ±σ and ±2σ bounds, to improve visual distinction. Also, we move the legend to the top and add corresponding explanations in the caption. In addition, we correct the issue where the axis ranges were not properly set, which caused the circle representing the Samalas eruption to be missing in some subplots.

As further suggested in the following comment 3) on marking the Pinatubo eruption in Figure 5, we include the temperature anomalies calculated from the 20th Century Reanalysis version 3 (20CRv3), marked with a green star in all subplots, to serve as a reference for the Pinatubo eruption. Because the Pinatubo eruption is not covered in the PMIP last millennium simulations (the main focus of this manuscript) but in the historical simulations (which have already been discussed in detail in other model comparison studies), we think that employing reanalysis data could provide a more robust reference here. This can also reveal the model biases mentioned

in previous studies (e.g., Driscoll et al., 2012) that CMIP models tend to overestimate the tropical cooling following tropical eruptions. We also add this point in our revised manuscript:

(Line 265-270) Also, the models tend to overestimate the tropical cooling compared with the anomalies after the 1991 Pinatubo eruption in the 20th Century Reanalysis version 3 (20CRv3, Slivinski et al., 2019), a bias also reported in CMIP5 historical simulations (Driscoll et al., 2012).

Our revised version of Figure 5 is shown below.

[Figure]

**Figure 5**. Near-surface air temperature (T2m, unit: °C) anomalies during the first winter after each eruption in PMIP last millennium simulations. Each subplot shows the T2m anomalies over 0°–30°N and northern Eurasia (55°N–70°N, 10°E–120°E) relative to the mean value over the last millennium. Tropical (TROP) eruptions are shown as red circles, and Northern Hemisphere extratropical (NHET) eruptions as blue circles. The sizes of the circles represent the magnitudes of the eruptions which are normalized to the Samalas eruption (the filled red circle in each subplot). The filled green star denotes the Pinatubo eruption based on the 20th Century Reanalysis version 3 (20CRv3). Dashed lines indicate the ±σ and ±2σ ranges of T2m anomalies over northern Eurasia during the last millennium.

*3) One of the previous hypotheses is that there is no NAO response or Eurasian temperature response. This study retorts that the choice of model and forcing data is, or can be, important, and that the signal may have been smeared out by different model responses and differences in the forcing data. One way to contribute to clarifying this question is to look at a one-model simulation ensemble. If the MPI-ESM-P model, according to the authors, is one of the more realistic models in this regard, would it be possible to examine the ensembles of historical simulations with this model for the Pinatubo eruption? What is the spread in that ensemble? Marking the Pinatubo eruption (perhaps with green colour) in the panels of Figure 5 would also help.*

**Response:**

Thank you for this suggestion. Our results indicate that the MPI-ESM1-2-LR model, with the latest PMIP4 volcanic forcing, demonstrates overall good performance. This model has 50

ensemble members from its CMIP6 historical experiment (1840–2014), which allows us to examine whether it can reproduce winter Eurasian warming after the Pinatubo eruption and to assess the spread across ensemble members. We also examine the 1883 Krakatau eruption, as previous studies (e.g., Bittner et al., 2016; Zambri and Robock, 2016) have suggested that CMIP5 models are able to capture winter warming when only consider these two largest tropical eruptions after 1850. To further support our findings, we add a new figure (Figure 8, shown below) in Section 3.4 together with the following text:

(Line 360-375) Overall, the MPI-ESM1-2-LR model demonstrates skill in capturing volcanic impacts on the winter North Atlantic climate in its last millennium simulation with the PMIP4 volcanic forcing, particularly after TROP eruptions. To further assess its performance, we analyse 50 ensemble members from its historical simulation (1850–2014) with CMIP6 forcing to examine post-volcanic SLP and T2m changes. This evaluation focuses on the two largest TROP eruptions since 1850 (the 1991 Pinatubo and 1883 Krakatau), as previous studies have shown that CMIP5 models can reproduce a strengthened Northern Hemisphere polar vortex and associated winter warming when only considering these two events (Bittner et al.,2016; Zambri and Robock, 2016). The ensemble means of the MPI-ESM1-2-LR historical runs exhibit a clear positive NAO pattern and Eurasian warming after both eruptions (Fig. 8). Nearly all members simulate post-volcanic tropical cooling signal. However, due to the strong natural variability of the Northern Hemisphere winter climate, as previously shown in Figure 3, the T2m anomalies over northern Eurasia display a large spread across the 50 ensemble members. Despite this wide spread, more members show Eurasian warming than cooling, and 17 members after Pinatubo and 13 members after Krakatau exhibit a warming signal that exceeds one standard deviation of their temperature variability. These results further indicate that, although a large spread exists, the MPI-ESM1-2-LR model performs well in reproducing post-volcanic wintertime climate responses.

[Figure]

**Figure 8**. (a) Sea level pressure (SLP) and (b) near-surface air temperature (T2m) anomalies in the 50-member ensemble mean of the MPI-ESM1-2-LR historical simulation after the 1991 Pinatubo eruption. White dots indicate regions where more than 30 of 50 members agree on the sign of the anomalies. (c) T2m anomalies over 0°–30°N and northern Eurasia (55°N–70°N,

10°E–120°E) relative to the mean value of the historical simulation period (1850–2014), with red circles denoting individual members and the blue diamond denoting the ensemble mean. T2m anomalies over northern Eurasia exceeding one standard deviation (σ) are filled with color. Panels (d–f) are the same as (a–c) but for the 1883 Krakatau eruption.

*4) The resolution of the figures needs to be improved. Many of them are multiple panels of relatively small size. A finer resolution would help the reader*

**Response:**

Thank you for pointing out this issue. We have replaced all figures with higher-resolution versions in our revised manuscript.

**Reply to reviewer 2:**

*Understanding the differing responses to tropical and Northern Hemisphere volcanic forcing is crucial. This knowledge not only aids us in making informed predictions regarding potential future eruptions but also serves as a valuable analogy for stratospheric aerosol injection. This paper provides a thorough comparison between multiple reconstructions and simulations. The consistency observed in both reconstructions and simulations reveals positive NAO anomalies and Eurasian warming following tropical volcanic eruptions. However, significant discrepancies emerge in the case of Northern Hemisphere eruptions. These findings are intriguing, and I believe this paper could be published in Climate of the Past following major revisions.*

**Response:**

Thank you for the positive comment on our manuscript. Below, we provide our point-by-point responses.

*My primary concern lies in the definition of the eruption year. The impact of a volcanic eruption on climate is contingent upon when and where its stratospheric aerosols evolve. Specifically, tropical to mid-latitude lower stratospheric aerosols absorb shortwave radiation, leading to warming, which in turn increases the meridional temperature gradient and enhances the polar vortex. The role of ozone deflection is not adequately represented in these PMIP models when explaining the enhanced polar vortex following tropical eruptions. For instance, consider the Samalas eruption, which is identified as occurring in 1258 in the GRA dataset, while it is recorded as 1257 in both the CEA and eVolv2k datasets. This discrepancy means that comparisons made for the winter NAO refer to the 58/59 winter in GRA, whereas they refer to the 57/58 winter in CEA and eVolv2k. In Fig. 4c, we can see that the middle-latitude lower tropospheric aerosol forcing is significantly large in GRA six months prior to the 58/59 winter, while it only appears one or two months before the 57/58 winter in CEA and eVolv2k. This suggests that the aerosol warming effect did not have sufficient time to exert its influence in the latter case. Moreover, for many historical eruptions, pinpointing the exact eruption month can be challenging. Therefore, I recommend defining the eruption year based on the maximum annual aerosol production in simulations and making subsequent comparisons accordingly.*

**Response:**

Thank you for raising this important point and for suggesting a revised definition of the eruption year. We agree that the timing of maximum aerosol forcing provides a more robust criterion for identifying eruption events, and we have therefore re-examined our analysis accordingly.

Using this revised definition, we find that the main differences compared to our original analysis appear in the superposed epoch analysis (Figure 3) and in the interpretation of individual events (Figure 5). For example, in the updated Figure 3 (attached below), the significant responses occur in year 0 relative to peak forcing year (the year of maximum annual aerosol production), corresponding to the first DJF winter of that year. Specifically, we note that most events reach their maximum aerosol production in the year following the onset of the

eruption. For those few events peaking in the same year as the onset, the forcing data are typically assigned with onset in January, which requires shifting the reference to the following winter to ensure sufficient time for their climatic influence to develop. Consequently, the eruption year +1 coincides with the maximum forcing year in most cases. This explains why the composite anomalies and corresponding spatial patterns of all selected eruptions (e.g., Figures 1, 2, 6, and 7, not shown here in our reply) remain nearly unchanged compared with the original version.

Therefore, to incorporate the suggestion of this comment without making the manuscript overly complex, we propose to retain Table 2 showing the starting year of the eruption for supporting the comparison in Figure 4. In the meanwhile, we add a supplementary Table S1 listing the year of maximum forcing used for other analysis. We also update the Methods section to clarify this definition and revise the relevant figures accordingly:

(Line 140-145) Similar to Liu et al. (2022), we identify the year of peak stratospheric sulfate aerosol loading as the key year for each eruption (Table S1) to ensure sufficient time for the volcanic aerosols to influence the subsequent winter climate. We use an 11-year time window with five years before and five years after the key year of each eruption and estimate statistical significance at the 95% confidence level by randomly resampling 10,000 sets of pseudo-events from the data (Brad Adams et al., 2003). Year 0 in the superposed epoch analysis denotes the first DJF winter with January–February in the peak forcing year, which is defined as the first winter following the eruption.

We hope this will address your concern while keeping the manuscript concise.

[Figure]

**Figure 3.** Winter near-surface air temperature responses to TROP and NHET eruptions in the last millennium simulations. (a) Time series of winter near-surface air temperature (T2m, unit: °C) anomalies over the tropical region (0°–30°N) for multi-model means forced by GRA, CEA, eVolv2k datasets. Grey shading denotes the spread among individual model simulations.

Red and blue triangles indicate TROP and NHET eruptions with magnitudes greater than the 1991 Pinatubo eruption defined in the eVolv2k dataset, respectively. (b-c) Results of superposed epoch analysis for tropical T2m over 0°–30°N after TROP and NHET eruptions, with data statistically significant at the 95% confidence level marked with asterisk. (d-f) T2m anomalies over 0°–30°N in the first winter following each eruption with respect to the mean value over the last millennium. The solid lines represent the linear regression of T2m anomalies against the magnitude of eruptions, with star symbols indicating that linear trends are statistically significant at the 95% confidence level using F-test. (g-l) are as (a-f) but for the regional T2m over northern Eurasia (55°N–70°N, 10°E–120°E).

*Abstract: The results of this paper are not sufficiently summarized and emphasized in the abstract. It is crucial to clarify the nature of the NAO response observed following tropical and Northern Hemisphere eruptions, as well as to outline the consistencies and discrepancies between the models and reconstructions. We can confidently conclude that tropical eruptions tend to enhance the NAO positive phase. However, there is no consistent conclusion regarding the effects of Northern Hemisphere eruptions.*

**Response:**

Thank you for bringing it up. We agree that the abstract can be improved. We revise it as follows:

**Abstract.** Large tropical (TROP) volcanic eruptions can influence North Atlantic climate by inducing a positive shift of the North Atlantic Oscillation (NAO), typically resulting in winter warming across northern Eurasia. However, these changes remain highly uncertain, as they may coincide with strong internal variability in Northern Hemisphere wintertime climate. In contrast, Northern Hemisphere extratropical (NHET) eruptions are proposed to have opposite impacts, but they have been comparatively less studied, and large uncertainties remain regarding the ability of climate models to capture volcanic responses. This study examines winter North Atlantic climate responses to TROP and NHET eruptions by comparing temperature and atmospheric circulation patterns from last millennium simulations with multiple proxy-based reconstructions. We find distinct differences in NAO-related climate changes in reconstructions, with TROP eruptions followed by a shift towards positive NAO and NHET eruptions associated with a negative NAO. In comparison, modelled responses exhibit a wide spread with strong dependence on the choice of volcanic forcing dataset. Notably, simulations using the latest volcanic forcing data show improved agreement with reconstructions, particularly for TROP eruptions. This model-proxy agreement provides a useful basis for investigating the mechanisms that drive positive NAO responses after TROP eruptions. However, the simulated impacts of NHET eruptions are less consistent and remain unclear. These results highlight the importance of improved volcanic forcing datasets, refined paleoclimate reconstructions, and robust statistical approaches to better constrain uncertainties in assessing volcanic impacts on North Atlantic climate.

*Section 2.2: To compare reconstructions and simulations, the criteria for event selection were similar to those used in previous work (Liu et al. 2022 Nature communications).*

**Response:**

We include this point in Section 2.2:

(Line 115) Similar to previous model-proxy comparison work on volcanic impacts (Liu et al., 2022), we select eruptions in the PMIP last millennium simulations according to the volcanic aerosol datasets used to force the models, as three different datasets are employed across the

15 simulations. […] For the proxy-based reconstructions, eruptions are identified from the volcanic forcing reconstruction of Sigl et al. (2015), which is derived from Greenland and Antarctic ice cores […]

*Lines 245: The composite aerosol forcing for these three datasets, as shown in Figs. 4c and 4d, is valuable for understanding the differing responses observed in the simulations.*

**Response:**

Thank you for bringing it up. We add more descriptions about Figs. 4c and 4d in this paragraph to clarify the differences in the representation of the Samalas and Laki eruptions across the three datasets, with particular attention to the aspects highlighted in your primary concern.

(Line 255-260) The Samalas eruption is identified as occurring in 1258 in the GRA dataset, where the mid-latitude aerosol forcing becomes significantly large about half a year before the following 1258/1259 DJF winter. In contrast, both the CEA and eVolv2k datasets assign the onset of the eruption to 1257. In particular, volcanic aerosol forcing in the CEA dataset begins only about two months before the 1257/1258 DJF winter, with its maximum forcing occurring afterward. This timing suggests that the aerosols may not have sufficient time to influence the climate during the first winter following the eruption, which could further complicate the comparisons among models forced by different datasets.

We also add the following text regarding this point in Discussion Section 4.3:

(Line 450-455) […] The temporal evolution and spatial distribution of volcanic forcing for the selected eruptions differ markedly among the GRA, CEA, and eVolv2k datasets (Fig. 4), and these differences are critical because the climatic impact of an eruption depends on when and where its stratospheric aerosols evolve. For many eruptions without a recorded eruption month, the events are defined as occurring in April in the GRA dataset, but in January in the CEA and eVolv2k datasets. Such discrepancies in the timing and duration of peak forcing complicate the comparison of winter signals, as it becomes difficult to ensure both sufficient time for the eruptions to exert their impacts and an accurate capture of the peak climate anomalies. This challenge is particularly pronounced for NHET eruptions, which are often less well documented and identified than TROP eruptions due to their smaller magnitudes and more geographically restricted impacts on the Northern Hemisphere. As a result, the simulated impacts of NHET eruptions are less consistent across models compared to those of TROP eruptions […]